# Chiral supramolecular assembly to enhance the magneto-optical rotation of organic materials

Leo Delage-Laurin[1,10], David Reger [2,3,10], Abdusalom A. Suleymanov [1,10], Zachary Nelson [1], Louis Minion[4,5,6], Steven E. Kooi [7], Jochen R. Brandt [8], Giuliano Siligardi [6], Robert P. Cameron[9], Jessica Wade [4,5] ✉, Timothy M. Swager [1] ✉ & Matthew J. Fuchter [2,3,5] ✉

Connections between magnetic field induced optical activity and chirality have a rich and complicated history. Although the broken inversion symmetry of chiral molecules generates 'natural' optical activity, magnetic optical activity is generated by breaking time reversal symmetry. Therefore, molecular chirality is not expected to influence magnetic optical phenomena, such as Faraday rotation. Here we show that the chiral supramolecular assembly of polymers can result in large Faraday effects (Verdet constants = $10^5$ °T$^{-1}$m$^{-1}$). This strong Faraday rotation, which is amongst the highest value known for organic materials, originates from the so-called Faraday B term. Typically, B term Faraday responses are weak. We demonstrate large amplification through excitonic coupling within the supramolecular assembly, where the chirality of the system controls the assembly formed. These observations provide an alternative means to enhance the Faraday rotation of low symmetry systems and clarify the role of chirality in previous reported materials.

The realisation of low cost, flexible magneto-optical (MO) materials that can operate at room temperature will transform multiple technological applications, from low-power computation and memory devices to safety testing in aviation and medical imaging[1–7]. Measurements of the optical rotation ($\theta_F$) of plane-polarised light as it passes through a material with a large Verdet constant ($V$) in an applied magnetic field ($B_z$) – so-called Faraday rotation (Equation 1) – can provide precise information about the applied magnetic field strength (Fig. 1a). $B_Z$ is the z-component of the magnetic field, which is parallel to the direction of light propagation, and $l$ is the thickness of the material.

$$\theta_F = V B_z l \tag{1}$$

The Verdet constants of commercially available MO materials, for example terbium gallium garnet (TGG), can reach -10$^4$ °T$^{-1}$m$^{-1}$ [8]. Such materials are traditionally based on inorganic ferro- and paramagnetic rare-earth crystals. It is difficult to integrate these crystals into new

[1]Department of Chemistry, Massachusetts Institute of Technology, Cambridge, MA, USA. [2]Department of Chemistry, Molecular Science Research Hub, Imperial College London, London, UK. [3]Department of Chemistry, Chemistry Research Laboratory, University of Oxford, Oxford, UK. [4]Department of Materials, Imperial College London, London, UK. [5]Centre for Processable Electronics, Imperial College London, London, UK. [6]Diamond Light Source Ltd, Harwell Science and Innovation Campus, Didcot, Oxfordshire, UK. [7]Institute for Soldier Nanotechnologies, Massachusetts Institute of Technology, Cambridge, MA, USA. [8]Department of Chemistry, Queen Mary University of London, London, UK. [9]SUPA and Department of Physics, University of Strathclyde, Glasgow, UK. [10]These authors contributed equally: Leo Delage-Laurin, David Reger, Abdusalom A. Suleymanov. ✉e-mail: jessica.wade@imperial.ac.uk; tswager@mit.edu; matthew.fuchter@chem.ox.ac.uk

**Fig. 1 | Measurements and materials used in this work. a** Illustration of the Faraday effect as it might appear in a simple experiment. Plane of polarisation (here represented by the plane of the electric field E) of a beam of light rotates about an angle $\phi_F$ as it passes through a material of thickness l in the presence of a parallel magnetic field $B_Z$. V is a constant of proportionality known as the Verdet constant and its value and sign are intrinsic to every material at each wavelength. **b** Chemical structures of **F8BT** and **[P]**- and **[M]**-**aza[6]H**, **c** Proposed structure adopted by thermally annealed thin films of **F8BT** blended with **aza[6]H** additives[28], **d** Normalised absorbance and CD for as-cast and annealed **F8BT:10 wt% aza[6]H** blends.

device formats due to their lack of mechanical flexibility, limited crystal scalability/size, and incompatibility with multifunctional smart technologies. Furthermore, such inorganic materials often require energy intensive cryogenic temperatures to generate a large MO response[9,10]. As an alternative, π-conjugated organic materials have recently been found to generate Faraday rotations with Verdet constants that rival the established inorganic MO-materials[11]. A comprehensive understanding of all the mechanisms that generate MO activity (MOA) in organic systems is yet to materialise, and new insights are needed for the design and optimisation of next-generation Faraday rotators.

The Faraday response of a material can be assigned to the Faraday A, B, and C terms, which describe distinct quantum-mechanical mechanisms for MOA that are most intense near regions of optical absorption[12]. Although there are detailed descriptions elsewhere[11], a brief discussion of the A, B, and C terms is instructive when designing MO active organic materials. In the presence of a magnetic field, the A term arises from the Zeeman splitting of degenerate ground or excited electronic states resulting in unequal optical frequencies and requires high symmetry organic molecules (for example extensively studied porphyrins)[13–22]. The C term is the magnetic field induced Zeeman splitting of degenerate ground electronic states and the unequal thermal populations of the resulting states, which requires high symmetry paramagnetic systems. Such stringent requirements of both the A and C terms restricts the potential of diamagnetic, low symmetry systems that are far more common amongst π-conjugated organic materials. The B term occurs due to magnetic field induced mixing of electronic states and is the main origin of MOA for lower symmetry materials, but is typically very small compared to the other terms

(A:B:C of 20:1:50)[23]. Methods to amplify B term Faraday rotation to practically useful levels are lacking.

It has been proposed that chirality enhances the MOA of organic materials. Thermally annealed thin films of polyfluorene with chiral sidechains have been reported to display $V = -38 \times 10^4 \, °T^{-1}m^{-1}$, whilst no rotation was observed in the absence of chiral sidechains[24]. Meanwhile, thin films of polythiophenes having chiral sidechains and helical microstructures also display significant MOA ($V = 7.6 \times 10^4 \, °T^{-1}m^{-1}$), and the sign of the rotation was shown to correlate with the absolute stereochemistry of the side chain[25]. The precise origins of the MOA in these materials is unclear, although molecular conformation has been cited to play a key role. The search for a connection between molecular chirality and Faraday rotation dates back to investigations on magnetism and Natural Optical Activity (NOA) of molecular crystals by Louis Pasteur[26]. Theoretical consensus has remained that chirality should have no influence on Faraday rotation, playing no role in the Faraday A, B, and C terms[27]. Barron and others have established that the origin of NOA is associated with spatial dissymmetry ("true" chirality), whereas the origin of MOA is associated with time reversal dissymmetry ("false" chirality)[26]. This poses many questions in the measured Faraday responses of chiral organic polymeric materials to date.

In this work, we report that chiral supramolecular assemblies of polymers can produce large Faraday effects. The significant Faraday rotation observed originates from the Faraday B term, a contribution that is typically weak. In this case, significant amplification arises from excitonic coupling within the supramolecular framework, with the handedness of the system directing the mode of assembly. These results establish a distinct approach for enhancing Faraday rotation in low-symmetry materials.

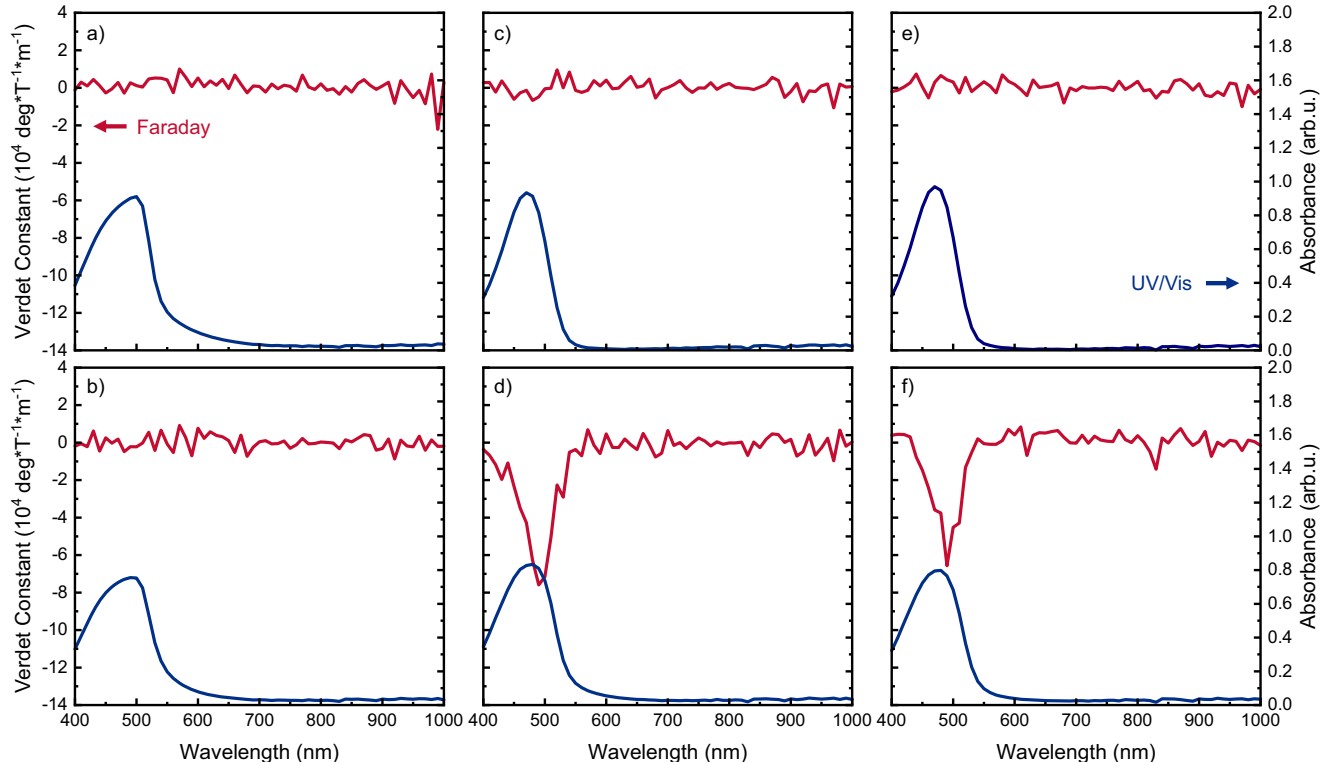

**Fig. 2 | Faraday rotation (red) and absorbance (blue) of polymer films. a** F8BT, thermally annealed; **b** F8BT:10 wt% racemic-aza[6]H, thermally annealed; **c** F8BT:10 wt% [*M*]-aza[6]H, as cast; **d** F8BT:10 wt% [*M*]-aza[6]H, thermally annealed; **e** F8BT:10 wt% [*P*]-aza[6]H, as cast; **f** F8BT:10 wt% [*P*]-aza[6]H, thermally annealed. All films have a thickness of ≈ 210 (±10) nm.

## Results and discussion

In an attempt to unravel the origins of Faraday rotation in chiral polymer thin films, we turned to a robust and controllable material system reported to show large NOA[28]. Specifically, we used blend thin films of the achiral polymer, poly(9,9-dioctylfluorene-alt-benzothiadiazole) (**F8BT**) and enantiopure chiral small molecule additive, [*M*]- and [*P*]-aza[6]helicene (**aza[6]H**) (Fig. 1b). The full origins of the large NOA of this material continue to be debated[29], but we have shown that when processed appropriately the large NOA arises from a helically twisted supramolecular assembly (Fig. 1c)[28]. In this assembly we observe excitonic coupling between adjacent polymer chains[30], and have proposed that higher order excitonic coupling terms (particularly those involving both the electric and magnetic transition dipoles) contribute to the large NOA[31]. Here we show that this same supramolecular assembly gives rise to very large Faraday rotation. Upon formation of a chiral phase in **F8BT:aza[6]H**, we observe robust and enantiomer *independent* MOA that is linearly dependent on the applied magnetic field. The Verdet constant achieved ($10^5 \, °T^{-1}m^{-1}$) is an order of magnitude larger than commercial materials, amplified through an excitonically enhanced response. As such, we conclusively demonstrate a strategy to enhance the MOA of low symmetry materials through supramolecular chemistry and clarify the role of chirality in the response.

We have extensively optimised and characterised the **F8BT:aza[6]H** system previously and shown that upon thermal annealing ($T = 140 \, °C$ for 10 min in a $N_2$ glovebox), this material displays strong NOA (>20,000 mdeg) in the lowest energy electronic transition, with the sign of the CD depending on the handedness of the enantiopure **aza[6]H** additive[28]. As such, these blend materials provide a very convenient means to study the role of chirality and supramolecular assembly compared to covalent (i.e., chiral sidechain) approaches: a single polymeric system can be studied in the non-assembled (as cast) or assembled (thermally annealed) state, with the handedness

of the helical assembly controlled by the handedness of the chiral additive. We measured the CD of freshly prepared samples, which only display large CD in the presence of enantiopure chiral **aza[6]H** additive and once the material has been thermally annealed (Fig. 1d), consistent with our previous work. Here we additionally characterised the chiroptical response of **F8BT** with racemic **aza[6]H** (Fig. 1d), which results in zero CD, likely due to ineffective assembly into appropriately-sized single handed domains.

Using these films, we proceeded to measure Faraday rotation using a previously reported home-built setup[13]. Our preliminary studies of enantiopure thermally annealed films gave curious absorption line shapes and an apparent change in sign of $\theta_F$ as a function of the handedness of the chiral additive. Further experimentation and theoretical analysis revealed that this outcome was caused by an artefact in our detector setup (see Supplementary Information). Ultimately, our original setup does not hold for materials with very large natural circular birefringence and miscalculates total light intensity. These observations are likely not unique to our data and highlight the need for caution when analysing MOA in chiral thin films with large NOA[32].

We adapted our experimental set-up to account for this issue and remeasured the Faraday rotation. The artefact-free spectra reveal one intense peak at ~500 nm, centred at the onset of the lowest energy electronic band (Fig. 2). Faraday rotation is negligible or very weak for neat achiral polymer films, **F8BT:racemic-aza[6]H** blends and non-annealed **F8BT:as-cast aza[6]H** blends (note, our set-up can only detect $V > 10^4 \, °/Tm$). Importantly, while left- and right-handed **F8BT:aza[6]H** blends (i.e., opposite helically twisted supramolecular assemblies) show opposite signs in their NOA (Fig. 1d), both enantiomers show the same sign in Faraday rotation (Fig. 2d, f). It occurs at the same wavelength and same magnitude for left- and right-handed **F8BT:aza[6]H** blends. This outcome is consistent with conventional understanding of MOA and its relationship with chirality (see above)

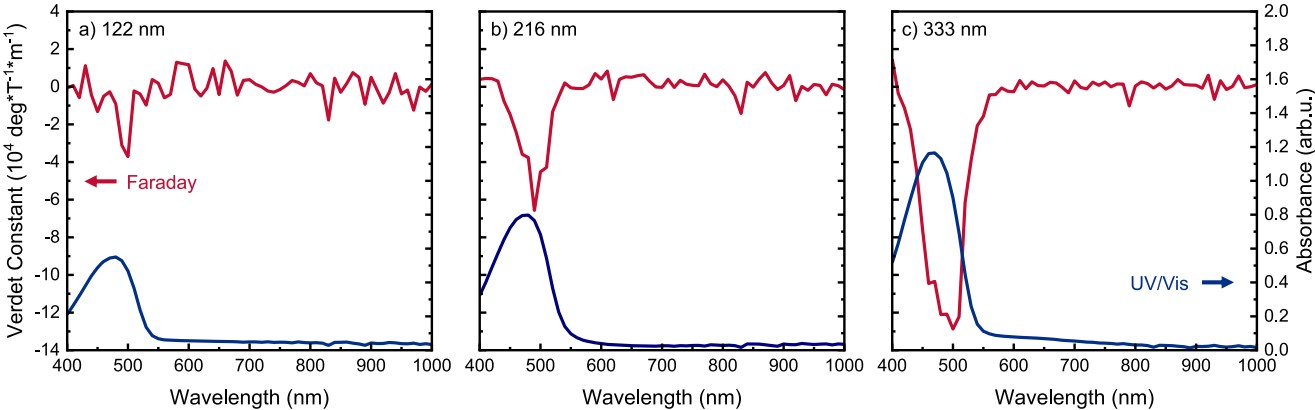

**Fig. 3 | Faraday rotation (red) and absorbance (blue) of polymer films as a function of thickness and field strength.** Faraday rotation of **F8BT:10 wt% [P]-aza[6]H**, thermally annealed with different thicknesses: **a** 122 nm; **b** 216 nm; **c** 333 nm.

and may indicate that prior reports of the Faraday rotation of chiral organics need to be revisited.

These data conclusively show that the large Faraday rotation originates from the chiral supramolecular assembly. Verdet spectra were also acquired using the highly collimated light of the Diamond Light Source B23 beamline. Magnetic circular birefringence (MCB), which gives rise to Faraday rotation, was extracted from Mueller-matrix polarimetry performed in a positive and negative applied magnetic field ($B_z$ = 1.3 T) and corrected for contributions of the substrate[33]. Despite the very different set-up for measuring the same quantity, the results are consistent with significant Faraday rotation at ~500 nm (Supplementary Fig. 7). Furthermore, the Faraday response of these materials behaves as expected by conventional MOA theory (Eq. 1), with a linear dependence on the applied magnetic field ($B_z$, Supplementary Fig. 8).

Although the Verdet constant of a material (at a given wavelength, $\lambda$) is expected to be independent of film thickness (l), we observe a thickness dependence of V for films <350 nm (Fig. 3a–c and Supplementary Fig. 9). We note that thickness dependent V has been previously observed in liquid crystal assemblies[34]. The thickness dependence of **F8BT:aza[6]H** blends also modifies the gradient of Verdet constant vs $B_z$ (Supplementary Fig. 8). We considered potential origins of this thickness dependent MOA, and whether it could be due to measurement artefacts (see Section 2 of Supplementary Information). We find that our observations are not consistent with an artefact, but instead point toward supramolecular origins, where domain size and order play a key role. The film thicknesses considered do not exceed the dimensions of the helically twisted assembly we have previously assigned to this material (Fig. 1c, Supplementary Table 1)[28] and we hypothesise that the maximum $V$ can only achieved when the thickness reaches the full extent of the substructures along the optical axis.

Although absolute stereochemistry is not important for Faraday rotation in these materials (Fig. 2), the formation of a supramolecular chiral assembly is critical to generating MOA, and results in a Faraday response ($10^5\,°T^{-1}m^{-1}$) over an order of magnitude larger than commercial Faraday rotators. Upon further analysis (Fig. 4), the spectrum suggests that the large Faraday rotation might originate from exciton coupling between adjacent polymer chains. This proposed mechanism seems reasonable given that the large NOA can also be attributed to excitonic coupling[28,31]. The MOA includes contributions from excitonically coupled dimers, which generate B term-like responses close in energy to one another and combine to form the A term-like line shape observed here. This so-called pseudo-A term is well known from related magneto circular dichroism (MCD) spectroscopy[16].

Applying the well-established theory of Faraday rotation to a simple model of an excitonically coupled dimer (Fig. 4), we find that the Verdet spectrum due to the lowest energy electronic transitions $j_\pm \leftarrow n$ (with other transitions ignored) has the form

$$V_j = -\frac{\mu_0 c N}{3\hbar}\omega^2\left(f_{j_+}B_{j_+} + f_{j_-}B_{j_-}\right) \quad (2)$$

with

$$f_{j_\pm} = \frac{\omega_{j_\pm n}^2 - \omega^2}{\left(\omega_{j_\pm n}^2 - \omega^2\right)^2 + \omega^2\Gamma_{j_\pm}^2} \quad (3)$$

where $|n\rangle$ is the ground state of the dimer, $|j_\pm\rangle$ are the first excited states of the dimer (together forming an exciton doublet), $N$ is the number density of dimers, $\omega = 2\pi c/\lambda$ is the optical angular frequency, $f_{j_\pm}$ are dispersive lineshapes, $B_{j_\pm}$ are Faraday B coefficients, $\omega_{j_\pm n}$ are transition angular frequencies, $\Gamma_{j_\pm}$ are linewidths and $\mu_O$ is the magnetic constant[12,23,27,35,36]. For suitable parameter choices, the variation of $V_j$ with $\lambda$ can take the form of a single negative feature (Fig. 4b), in qualitative agreement with the observed Verdet spectrum. A closer fit can be obtained by accounting for additional transitions ($k_\pm \leftarrow n$ etc.), however our measurements do not extend to sufficiently low $\lambda$ to enable an explicit comparison. We note that for both left- and right-handed systems (Fig. 4c, d) the fit to our experimental data is excellent.

In this study we explore the role of chirality in the MOA of molecular materials. We show that the absolute stereochemistry/handedness of the material has no effect on the sign of the Faraday rotation. Instead, we find that chirality drives the polymer self-assembly into a (single handed) helically twisted phase, where strong excitonic coupling between adjacent polymer chains results in a large amplification of the Faraday B term. Given that most diamagnetic low symmetry organic materials cannot achieve MOA via other mechanisms (i.e., A or C term), supramolecular assembly offers a strategy for enhancement of Faraday rotators. Chirality is a key opportunity to control pathway complexity in such assembly processes[37] and therefore will likely remain a useful design criteria for future MOA materials. Even without further optimisation, it is apparent that chiral polyfluorene blend materials have a Faraday response that significantly exceeds that of commercial inorganic crystals. Although these systems may not be appropriate for optical isolators, due to their large natural optical activity and high absorption, their high Faraday responses and flexibility open the door to new applications in low power magnetometry. This advocates the study and application of chiral molecular systems in low-cost, flexible magnetic field sensors.

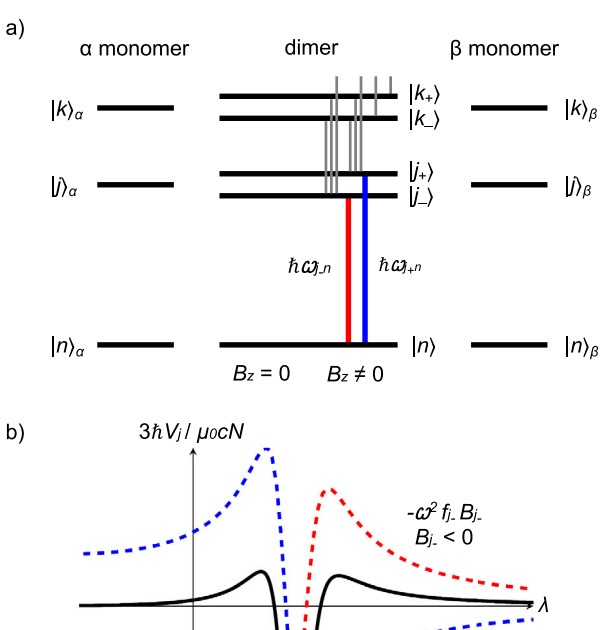

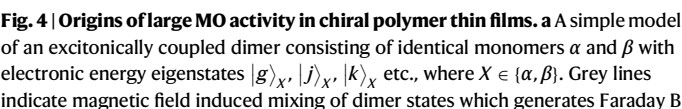

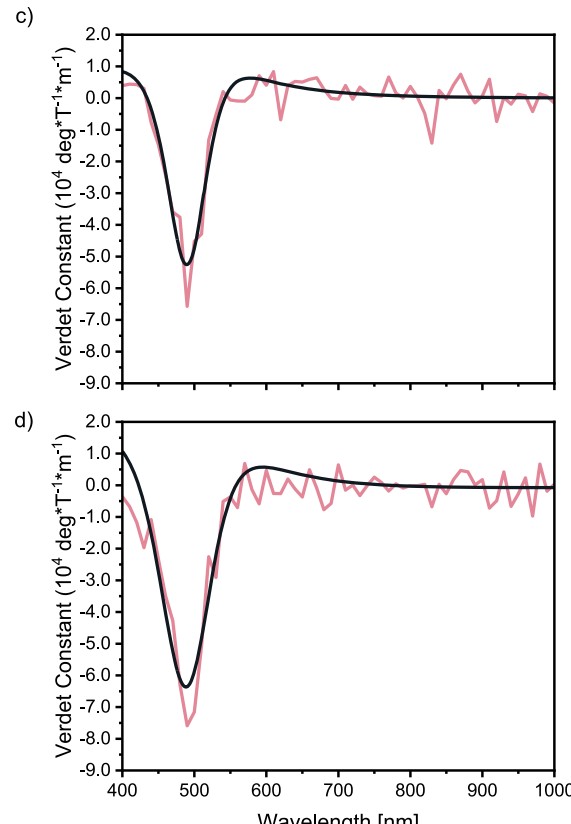

**Fig. 4 | Origins of large MO activity in chiral polymer thin films. a** A simple model of an excitonically coupled dimer consisting of identical monomers $\alpha$ and $\beta$ with electronic energy eigenstates $|g\rangle_X$, $|j\rangle_X$, $|k\rangle_X$ etc., where $X \in \{\alpha, \beta\}$. Grey lines indicate magnetic field induced mixing of dimer states which generates Faraday B

terms. **b** The Verdet spectrum of the lowest energy electronic transition $j_\pm \leftarrow n$ (with other transitions ignored) can take the form of a single negative feature. **c** Least squares fit (black) to the experimentally observed (pale red) Faraday rotation data for the 216 nm thick film of **F8BT: [M]-aza[6]H** and **d** F8BT: [P]-aza[6]H.

## Methods

### Materials
**Aza[6]H** was synthesised as previously reported[38]. It was separated into the enantiomers by Reach Separations, UK via chiral HPLC. Before use the **aza[6]H** enantiomers were further purified via recycling GPC to remove other small impurities. **F8BT** was provided by Cambridge Display Technology, UK (company number: 02672530) and used without further purification.

### Sample preparation
The achiral conjugated polymers **F8BT** (MW = 31 kDa) or and **aza[6]H** were dissolved in toluene to a concentration of 35 mg/mL (**F8BT**) and blended to form a 10 wt% polymer: **aza[6]H** solution. Ultra-thin glass substrates (#1.5H, 170 μm, 22 × 22 mm) were sequentially rinsed in an ultrasonic bath in acetone and isopropyl alcohol for 15 min, which was repeated three times. The substrates were then transferred to an oxygen plasma asher for 5 min at 90 W before spin-coating. To achieve large-area, uniform, films the ultra-thin glass substrates were held on glass microscope slides using a drop of water by capillary forces. Thermal annealing (TA) took place at 140 °C for 10 min in a $N_2$ glovebox, with <0.1 ppm $H_2O$ and $O_2$.

### Characterisation methods
Faraday Rotation spectra were measured on a home-built instrument which is further described in Section 1 of the Supplementary Information. Faraday Rotation spectra were also measured using the Mueller-matrix polarimeter (MMP) at the B23 beamline at the Diamond Light Source synchrotron. The instrument is based on the design of Hussain et al.[39]. The sample was placed in the centre of a

1.3 T permanent magnet sample holder, and measurements were taken with the magnet field positive and negative. The circular birefringence was estimated from the Mueller matrices by (m12 + m21)/2m00 and the contribution from the substrate removed by subtracting measurements on a bare substrate. MCB was calculated as (CB( + B) − CB(-B))/2, where CB( +/-B) is the measurement in positive/negative magnetic fields. CD spectra were measured on a Chirascan Plus V100 spectrometer from Applied Photophysics. For films demonstrating strong chiroptical activity (>2000 mdeg), a correction factor is used to calculate the ΔA and accurate CDmdeg when using the benchtop Chirascan Plus CD instrument, but unnecessary with the B23 MMP. UV/Vis spectra were measured in parallel with the Faraday rotation on the same instrument or together with the CD-spectra on the same instrument. Film thicknesses were measured on a JEOL 6010LA Scanning Electron Microscope (SEM).

## Data availability
Additional data associated with this study can be found in the Supplementary Materials or are available from the corresponding authors upon request. Source data are provided with this paper.

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

## Acknowledgements

We acknowledge Diamond Light Source for time on Beamline B23 under Proposals SM29153-3, SM31975-1 and SM33533-2 (M.J.F., J.W., and G.S.) and Cambridge Display Technology Limited (company number 02672530) for providing the achiral polymers. L.M. would like to thank the EPSRC and SFI Centre for Doctoral Training in Advanced Characterisation of Materials (Grant Ref: EP/S023259/1) and Diamond Light Source Ltd. for joined funding a PhD studentship with Imperial College. This work was additionally funded by the EPSRC through grant number EP/R00188X/1 (M.J.F.). This material is based upon work supported by the Air Force Office of Scientific Research under award numbers FA9550-22-1-0051 (T.M.S.) and FA9550-23-1-0633 (M.J.F and J.W.). J.W. thanks the Royal Society (URF\R1\231732). D.R. thanks the German Research Foundation (DFG) for support via the postdoctoral Walter Benjamin Programme (project number 500635930) and the Imperial College Postdoc and Fellows Development Centre (PFDC) for travel funding. A.A.S. is grateful to the Swiss National Science Foundation for a Postdoc Mobility Fellowship (P500PN_202689). J.R.B. gratefully acknowledges the support of a Royal Society University Research Fellowship (URF \R1\201639). R.P.C. gratefully acknowledges the support of a Royal Society University Research Fellowship (URF\R1\191243).

## Author contributions

Z.N., T.M.S., J.W., and M.J.F. conceived the idea. L.D.-L., D.R., A.A.S., and Z.N. led the experimental study and performed the Faraday rotation measurements. L.M. performed the MCB experiments with assistance from G.S. S.E.K. modified the Faraday rotation instrument. J.W. prepared additional thin films. D.R. and J.R.B. helped with materials and sample preparation. R.P.C. performed the mathematical analysis and interpretation as well as contribute to wider data analysis. L.D.-L., D.R., A.A.S., L.M., R.P.C., T.M.S., J.W., and M.J.F. drafted the manuscript. All authors helped with data analysis. All authors contributed to the revision of the manuscript. T.M.S., J.W., and M.F. acquired funding and supervised the project.

## Competing interests

M.J.F. is an inventor on a patent concerning chiral polymeric materials (WO2014016611). The remaining authors declare no competing interests.
