## [Transparent Peer Review file · Nature Communications]

Chiral supramolecular assembly to enhance the magneto-optical rotation of organic materials.

Corresponding Author: Professor Matthew Fuchter

Version 0:

Reviewer comments:

Reviewer #1

(Remarks to the Author)

In this contribution the authors detail measurements of magnetic optical rotation in thin films of achiral polyfluorene type polymer doped with enantiomerically resolved helicene derivative.

The novelty here is 1) the large value of the Verdet constant and 2) the strong correlation between the magnitude of the magnetic optical rotation and the natural optical activity when varying the doping with helicene, the thermal annealing and the thickness of the films. This suggests a strong connection between the helical arrangement of the polymer chains in the film and the magnetic optical activity.

This report is new and highly interesting. Yet the following issues are not fully clear to me.

1) The high values of the Verdet constant describing the magnitude of the magnetic optical rotation occur in films where also the natural optical activity is very large. From previous research we know that the circular dichroism, i.e. the difference in absorption of left and right circularly polarized light, reaches very high values with the degree of circular polarization in the transmitted light going close to the maximum ± 2 .

This high circular component in the light transmitted through the films may however have significant implications for practical application of the effects reported here.

Magnetic optical rotation is commonly used to build optical Faraday isolators, i.e. optical elements that block the transmission in one direction but transmit the light in the other direction. To block the unwanted beam one uses normally a linear polarizer. Now in the case where the light transmitted through the film is highly circularly polarized due to the natural optical activity, the magnetic optical rotation can still be very large. Yet from an application perspective the high magnetic optical rotation is not that useful anymore because the transmitted light has also become highly circularly polarized and can no longer be blocked selectively with a linear polarizer. Note also that in case of purely circularly polarized transmitted light, optical rotation either of natural or magnetic origin, can no longer be defined.

To give the readers a way to assess this potential issue with practical applicability, I suggest that the authors also detail how large the circular dichroism in their film actually is. Can the authors show the circular dichroism of their film to give the readers an objective and complete measure of the polarization state of the light after transmission through the film?

2) The Verdet constant reaches very high values for wavelengths that overlap with the absorption band, implying that light not only gets rotated magnetically but also gets absorbed. In practical application one is usually most interested in wavelength regions where losses due to absorption are minimal. In this perspective, can the authors estimate how large the absorption losses will be in films where the optical rotation induced by moderate magnetic field strength reaches 90 degrees? Rotation of an angle of 90° is often desired in applications. A 'new' figure of merit combining the Verdet constant and the absorbance may be very insightful for the readers.

3) The manuscript is not very explicit about the supramolecular organization that gives rise to the optical effect observed here. In the SI the authors mention for instance some a "blue-Phase cylinder spacing" of about 300 nm, which seems most relevant. This needs some more explanation.

Reviewer #2

(Remarks to the Author)

The authors describe very efficient Faraday rotation in thin films of poly(9,9-dioctylfluorene-*alt*-benzothiadiazole) doped with helicene. All experiments are conducted with great care to avoid any possible artifacts. Their conclusion is that supramolecular assembly offers a strategy to enhance the Verdet constant. Furthermore, linking the observed Faraday rotation to the Faraday B-term is interesting.

However, I have the following concerns:

1) previous investigation of Faraday rotation in conjugated molecules and polymers also showed that supramolecular assembly is a crucial factor in the High Verdet constants that are observed in organic materials. See for example the following references:

J. Phys. Chem. C 2008, 112, 8032–8037

Macromol Rapid Commun. 2006;27:1920–1925

Furthermore, that chirality can be used to manipulate supramolecular assembly in organic conjugated systems is well-known. Hence, it is not surprising that chirality can be exploited to enhance the Faraday response which is highly dependent on the supramolecular organization. The authors should stress the novelty of their research as this is not clear from the manuscript.

2) It is true that the investigated materials have a very high Verdet constant, but not dramatically different from materials, achiral or chiral, that were previously investigated by other groups. Furthermore, the high Verdet constant is only observed under resonant conditions. Outside the absorption band, the Verdet constant is very low which makes it less applicable than the authors indicate.

3) linking the High Faraday response to the B-term is very interesting. However, this issue was already (although briefly) discussed in J. Am. Chem. Soc. 2022, 144, 11912–11926 by the same authors.

4) what about rigorous quantum mechanical calculations? This would have been interesting to verify the mechanism for high Faraday rotation in organic materials and verify conclusions made in the manuscript.

5) can the authors provide a clear structure-property relationship that may help us to design new organic materials for Faraday rotation. What type of molecular structure do we need (conjugation, aromaticity, ...)

conclusion: while the data presented in this manuscript are sound, it is unclear what the novelty is, how they can contribute to the design of novel materials, and whether these results will be interesting for a more general readership. I do not recommend publication. Other, more specialized journals may be more suitable.

Reviewer #3

(Remarks to the Author)

This report describes the magneto-optical activity of chiral supramolecular polymers. They prepare thermally annealed blends of an achiral polyfluorene and a chiral azahelicene, specifically aza[6]H. The results include CD, MCB, Faraday rotation, and absorption data of the pure compounds as well as the thermally-annealed blends. The primary motivation for this publication is the connection between natural optical activity (due to a chiral center) and the magneto-optical response of the material. At present, there is little theoretical connection between these two effects or phenomena. However, there are many in the magneto-optical community, the CISS community and the chiral polymer community that are beginning to acknowledge that there must be a connection. Indeed, the authors invoke an excitonic coupling mechanism to explain their results. The reasoning being that the coupling of electric dipoles must necessarily couple magnetic dipoles; magnetic dipoles associated with an electronic transition are directly measured in magneto-optical spectroscopy, regardless of the technique. Thus, this will be a cutting-edge report when suitable for publication. I have several comments below that must be addressed prior to publication.

1. As stated above, all magneto-optical spectra measure the magnetic dipole associated with an electronic transition. The magnetic dipole of an intense electronic transition is larger than that of a weak electronic transition. Figure 3 shows the absorption and Faraday rotation spectra as a function of thickness, but how does the absorption and Faraday rotation change as a function of concentration of the chiral center at a selected thickness, say 333 nm. Was this study performed? The Faraday rotation is equivalent to the MORD (Magneto-Optical Rotatory Dispersion) spectrum, which can be modeled by a simple second derivative of the absorption spectrum. Have the authors considered this simple calculation and overlaying it with the Faraday rotation? If the Faraday rotation spectral response and this second derivative deviate from one another as a function of concentration, then this might indicate a synergistic response, or at least other contributions to the Faraday rotation spectrum.

2. The authors clearly state that the thickness of the polymer is critical to the magnitude of the Faraday rotation. I believe that they are obligated to characterize the material more substantively. For example, how do the authors know that the effect is not due to some new liquid crystalline domain or similar phase? This is critical since the report cites a molecular (supramolecular) mechanism and not a material one. I recommend polarized microscopy or some XRD or X ray scattering or similar to help define the material more clearly; of course, other approaches are equally valid. If the sample has created some phase or domains with differing refractive indices, then the absorption will be different and accordingly the magnetic dipoles associated with those domains will be different. Or, thought another way, what is the annealing process doing and what processes are operative that lead to the increase in Faraday rotation as the sample thickness is increased?

3. The authors report the MCB (Magnetic Circular Birefringence) data and note that "Despite a very different measurement methodology, the obtained data is consistent with significant Faraday rotation..." Of course, the authors know that Magneto-Optical Rotary Dispersion, Magnetic Circular Birefringence and Faraday rotation all measure the same magnetic permeability of the sample. The way in which this section is written almost implies that these are different quantities. I urge the authors to write this in more clear fashion.

4. I congratulate the authors for addressing their instrumental artifacts. The balanced detection method is only appropriate for samples exhibiting small Faraday rotations; samples that feature large rotations will show artifacts due to signal processing errors. Some researchers have placed a quarter wave plate retarder after the sample and before the detector to compensate for this. I believe that more discussion of these instrumental details in the literature will help others avoid mistakes.

5. Magneto-optical activity can be measured by Faraday rotation, MCB, MORD and MCD (Magnetic Circular Dichroism). In fact, the total magneto-optical response of a material is the sum (Faraday rotation or equivalent and MCD), even though a device may operate on a Faraday rotation mechanism. The authors note the interest in porphyrins, but only cite their work (citations 13 and 14). MCD spectra of porphyrins is well studied (for decades and recently) and some other contributors should be listed. For example, the separate work of Stillman and Mack (though they have some collaborative manuscripts) have discussed in detail the magneto-optical properties of porphyrins that should be mentioned. But, one could equally cite Stephens (cited in other contexts), Gouterman, Djerassi, Michl, and many others. While their (collective) primary interest was/is band assignment and electronic structure, their interpretations are applicable to Faraday rotation.

6. Faraday B-terms typically show peak maxima (or minima, if negative) that coincide with the absorption maxima. In all of the spectra shown here, there appears to be a measurable shift to lower energy of the Faraday rotation peak minima (these are negative peaks) relative to the absorption maxima. Do the authors have insight into this observation? Is this due to some instrumental feature (does the Wollaston prism have a poor spectral response here), or is this due to the presence of a positive B-term on the high energy side that is smaller in magnitude?

Version 1:

Reviewer comments:

Reviewer #1

(Remarks to the Author)

Yes I am happy with the revisions

One minor remark: the revised version has some additional references among which

-Jacek, W., Josef, M. Perimeter model and magnetic circular dichroism of porphyrin analogs. *J. Org. Chem.* 56,2729-2735 (1991).

-Keegan, J.D., Bunnenberg, E., Carl, D. Magnetic circular dichroism studies - 63. Sign variation in the magnetic circular dichroism spectra of some perimeter symmetric metallo porphyrins. *Spectrochim. Acta A* 40,287-297 (1984)

I believe that the authors are referring to work by Josef Michl in the first reference and to work by Carl Djerassi in the second

Reviewer #3

(Remarks to the Author)

The response from the authors is satisfactory. The increased references are suitable though I would add

Stillman et al. *Journal of Photochemistry and Photobiology* 2018, 22, 1111

Mack *Chem. Rev.* 2017, 117, 3444

Turner et al *Inorg. Chem.* 2024, 63, 3630

Otherwise, I recommend acceptance of the manuscript as is.

REVIEWER COMMENTS

We are grateful to the Reviewers for their time and useful suggestions to improve our manuscript. We have attempted to respond to each of their comments below.

Reviewer #1 (Remarks to the Author):

In this contribution the authors detail measurements of magnetic optical rotation in thin films of achiral polyfluorene type polymer doped with enantiomerically resolved helicene derivative.

The novelty here is 1) the large value of the Verdet constant and 2) the strong correlation between the magnitude of the magnetic optical rotation and the natural optical activity when varying the doping with helicene, the thermal annealing and the thickness of the films. This suggest a strong connection between the helical arrangement of the polymer chains in the film and the magnetic optical activity.

This report is new and highly interesting. Yet the following issues are not fully clear to me.

We highly appreciate the enthusiasm of the reviewer and are grateful for the opportunity to address their questions.

1) The high values of the Verdet constant describing the magnitude of the magnetic optical rotation occur in films where also the natural optical activity is very large. From previous research we know that the circular dichroism, i.e. the difference in absorption of left and right circularly polarized light, reaches very high values with the degree of circular polarization in the transmitted light going close to the maximum ± 2 .

This high circular component in the light transmitted through the films may however have significant implication for practical application of the effects reported here.

Magnetic optical rotation is commonly used to built optical Faraday isolators, i.e. optical elements that block the transmission in one direction but transmit the light in the other direction. To block the unwanted beam one uses normally a linear polarizer.

Now in the case where the light transmitted through the film is highly circularly polarized due to the natural optical activity, the magnetic optical rotation can still be very large. Yet from an application perspective the high magnetic optical rotation is not that useful anymore because the transmitted light has also become highly circularly polarized and can no longer be blocked selectively with a linear polarizer. Note also that in case of purely circularly polarized transmitted light, optical rotation either of natural or magnetic origin, can no longer be defined.

To give the readers a way to asses this potential issue with practical applicability, I suggest that the authors also detail how large the circular dichroism in there film actually is. Can the authors show the circular dichroism of their film to give the readers a objective and complete measure of the polarization state of the light after transmission though the film ?

We appreciate the reviewer's detailed consideration of the interplay of natural and magnetic optical activity. The reviewer is correct: the large natural optical activity of these systems close to the region of highest magnetic optical activity precludes their application as optical isolators, where they would not be competitive with commercially available inorganic crystalline Faraday rotators. However, there are many other applications of Faraday rotators that these systems would be well suited for; including in low-power, conformable, ambient magnetic field sensing devices (referred to in the first sentence in the introduction as well as in references 1-7). For these applications the crucial parameters are high Verdet constants, flexibility and sufficient light transmission to enable measurement of optical rotation. From our perspective, an

increase in the ellipticity of the light, even with some losses due to absorbance (as long as there is still a detectable amount of light transmitted), should not compromise these applications.

The circular dichroism of the materials is reported in Figure 1d) and was the subject of a detailed previous study of our group: 10.1038/s41467-020-19951-y. Nonetheless, for the precise films used in this manuscript, the dissymmetry factor of absorption (g_{abs}) is $|\sim 0.8|$ for 220 nm thick films. In our revision, we have included additional measurements of optical rotation collected at the Diamond Light Source. By projecting a linear Stokes vector through the Mueller matrix recorded for the samples at different magnetic fields we can estimate the polarisation vector of the light exiting the sample. ($S_{out} = \underline{\mathbf{M}} \cdot S_0$).

This analysis has now been included in the Supplementary Information as Supplementary Figures S12 and S13 (shown below) including the raw experimental Mueller matrices as Figure S12 and S13 and the predicted polarisation ellipses as Figure S14. The following text was added to the as well:

9. Polarization of light after passing through the film

To establish the polarization that is induced into the linearly polarized light after passing through the chiral films we projected a linear Stokes vector through the Mueller matrix data (collected in transmission with the Mueller Matrix Polarimeter (MMP) at Diamond Light Source B23 beamline)⁶ recorded for samples at different magnetic fields. This allows us to estimate the polarization vector of the light exiting the sample. ($S_{out} = \underline{\mathbf{M}} \cdot S_0$).

Figure S11: Experimental transmission Mueller matrix of an F8BT:P-aza[6]helicene film recorded in a magnetic field of +1.3T.

F8BT:P-aza[6]helicene -1.3T

Figure S12: Experimental transmission Mueller matrix of an F8BT:P-aza[6]helicene film recorded in a magnetic field of -1.3T.

(P) – aza[6]helicene : F8BT

Figure S12: Predicted polarization ellipses of 500 nm light after transmission through a 220 nm thick sample of F8BT:P-aza[6]helicene in both a positive and negative magnetic field. The ellipses were predicted by multiplying a 45° linearly polarised Stokes vector ($S_0=[1,0,1,0]$) by the experimentally measured Mueller matrix for the sample at 500 nm ($S_{out} = \underline{M} \cdot S_0$). Ellipses were then visualised using the pypolar package for Python.⁷

These data show conclusively that while the light becomes circularly polarised after passing through the samples, there is a detectable and consistent shift in polarisation ellipse major axis

angle under a magnetic field. It is this change that further demonstrates sensitivity to magnetic fields and opens the door to potential applications in magnetometry.

In addition to adding this data to the SI, we have also adjusted the conclusion to the manuscript to put further emphasis on magnetic field detection applications:

Even without further optimisation, it is apparent that chiral polyfluorene blend materials have a Faraday response that significantly exceeds that of commercial inorganic crystals. Although these systems may not be appropriate for optical isolators, due to their large natural optical activity and high absorption, their high Faraday responses and flexibility open the door to new applications in low power magnetometry. This advocates the study and application of chiral molecular systems in low-cost, flexible magnetic field sensors.

2) The Verdet constant reaches very high values for wavelengths that overlap with the absorption band, implying that light not only gets rotated magnetically but also gets absorbed. In practical application one is usually most interested in wavelength regions where losses due to absorption are minimal. In this perspective, can the authors estimate how large the absorption losses will be in films where the optical rotation induced by moderate magnetic field strength reaches 90 degrees? Rotation of an angle of 90° is often desired in applications. A 'new' figure of merit combining the Verdet constant and the absorbance may be very insightful for the readers.

At this request, we have calculated the necessary film thickness for a 90° rotation of the angle (considering the Verdet constant of our 333 nm film and a field strength of 1 T) which is ~0.69 mm, which is considerably thinner than the inorganic crystals exploited in commercially available optical magnetometers (several cm). We would also however like to reiterate our response to Comment 1, absorption losses do not negate potential applications of such materials, so long as a detectable optical rotation is measurable. Our study suggests that through careful molecular design in the future (e.g. chiral systems with large Verdet constants and weak absorption) it should be possible to further expand the performance of organic materials.

At the request of the reviewer we have added a table (Table S2) with the maximum Figure of Merit (FoM) for each film to the supplementary information:

8. Figure of Merit (FoM) for Faraday rotation

To better evaluate the degree of rotation vs. the absorbance at a given wavelength, we determined the maximum FoM (Table S2) achieved for the investigated films. The FoM is defined as Faraday rotation in deg/T divided by the dimensionless absorbance A .¹ Please note that the wavelength of maximum FoM is not necessarily the wavelength of the maximum Verdet constant. For instance, the 333 nm thick F8BT: aza[6]H-P films show a broader magneto-optical response leading to a significant Verdet constant at the slope of the absorbance peak and therefore to a significantly increased maximum FoM.

Table S2: Maximum FoM for the investigated film.

Film	Wavelength of maximum FoM [nm]	Absorbance	Verdet constant x 10 ⁴ [°/Tm]	Figure of Merit [°/T]
F8BT: aza[6]H-P (122 nm)	500	0.47	-3.7	-0.010

F8BT: aza[6]H-P (216 nm)	490	0.77	-6.6	-0.019
F8BT: aza[6]H-M (216 nm)	500	0.74	-7.2	-0.021
F8BT: aza[6]H-P (333 nm)	510	0.69	-12.2	-0.059

3) The manuscript is not very explicit about the supramolecular organization that gives rise to the optical effect observed here. In the SI the authors mention for instance some a “blue-Phase cylinder spacing” of about 300 nm , which seems most relevant. This need some more explanation.

We apologise for not being more explicit about the supramolecular organisation. We have described the structure and chiroptical properties of this material extensively in previous work (10.1038/s41467-020-19951-y). In short, we propose the polymer assembles into a weakly ordered chiral, double twist cylinder-type blue phase, based on a combination of resonant soft X-ray scattering and high-resolution AFM, with the dimensions cited in the manuscript (e.g. approx. 300 nm). Some aspects of the supramolecular assembly are still uncertain (e.g., the distribution of the chiral additive in the blend system) owing to the lack of any long-range order and the complexity associated with generating a complete molecular model (further commented on below). To address this comment and help the reader better understand the materials system, we have added a brief summary of our prior work to the revised Supplementary Information (section 6):

The supramolecular structure within F8BT:aza[6]helicene films has previously been studied by our group.^[1] Specifically, using resonant X-ray scattering we observed an in-plane feature with periodicity on the order of 280nm, whilst high-resolution AFM revealed 25nm wide twisted fibrils. A combination of optical modelling and spectroscopic Mueller matrix ellipsometry ruled out the presence of single domain or multi-domain cholesteric-stack like organisations.^[1] The X-ray scattering results are consistent with a weakly ordered “blue-phase”, an arrangement in which cylinders formed from twisted polymer fibrils assemble perpendicular to one another with a periodicity of 280nm (as shown in Figure 1c).

Reviewer #2 (Remarks to the Author):

The authors describe very efficient Faraday rotation in thin films of poly(9,9-dioctylfluorene-alt- benzothiadiazole) doped with helicene. All experiments are conducted with great care to avoid any possible artifacts. Their conclusion is that surpamolecular assembly offers a strategy to enhance the Verdet constant. Furthermore, linking the observed Faraday rotation to the Faraday B-term is interesting.

We thank the reviewer for their supportive comments.

However, I have the following concerns:

1) previous investigation of Faraday rotation in conjugated molecules and polymers also showed that supramolecular assembly is a crucial factor in the High Verdet constants that are observed in organic materials. See for example the following references:

J. Phys. Chem. C 2008, 112, 8032–8037

Macromol Rapid Commun. 2006;27:1920–1925

Furthermore, that chirality can be used to manipulate supramolecular assembly in organic

conjugated systems is well-known. Hence, it is not surprising that chirality can be exploited to enhance the Faraday response which is highly dependent on the supramolecular organization. The authors should stress the novelty of their research as this is not clear from the manuscript.

We apologise if we did not make the significance of supramolecular assembly in achieving high Verdet constants sufficiently clear in our manuscript. Although there is indeed prior literature that suggests supramolecular assembly and/or chirality is useful in obtaining Faraday rotation, there are a number of limitations in the prior studies. For example, most studies do not consider both left, right-handed and racemic systems when considering chirality. Furthermore, we believe that prior assignments on the role of chirality are incorrect, likely due to artefacts in the measurements caused by detection geometries, large natural optical activity and/or other optical anisotropies. Given this, we believe a more fundamental understanding of *why* chiral supramolecular assembly enhances Faraday rotation has been lacking. We have attempted to address this in our manuscript. We summarise the key novelty of our study as follows:

We establish that prior reports on the influence of handedness, e.g., sign flip or varying extent of the Verdet constant from different enantiomers are incorrect, and suggest the reasons for these erroneous assignments.

We show that chiral supramolecular order can have a significant positive influence on the Faraday rotation, with robust comparator materials.

We deliver a mathematical model on how the B-term response in chiral polymer assemblies, caused by excitonically coupled chromophores, can significantly enhance the magneto-optical response.

We suggest that the combination of chiral supramolecular aggregation, particularly in systems with strong excitonic coupling, is a clear design strategy to access large Faraday rotation for lower symmetry organic materials. This is an important outcome for the field as A- and C-term Faraday rotation materials – which require high molecular symmetry – have been the focus to date.

2) It is true that the investigated materials have a very high Verdet constant, but not dramatically different from materials, achiral or chiral, that were previously investigated by other groups. Furthermore, the high Verdet constant is only observed under resonant conditions. Outside the absorption band, the Verdet constant is very low which makes it less applicable than the authors indicate.

We note that the reviewer has not given specific details of the prior studies to which they are referring so it is difficult to respond in specific terms. In our manuscript, we have tried to accurately benchmark our systems against prior literature, which are cited. We would be glad to consider the inclusion of other prior studies, should it be required. We would like to emphasise again that key to our manuscript is the understanding we generate on the response of these materials, rather than their large measured Verdet constant alone (see response to Reviewer #2, comment 1).

Regarding concerns on the overlap between Verdet constant and absorbance, please refer to the responses to comments 1) and 2) from Reviewer #1.

3) linking the High Faraday response to the B-term is very interesting. However, this issue was

already (although briefly) discussed in *J. Am. Chem. Soc.* 2022, 144, 11912–11926 by the same authors.

The paper cited by the reviewer is a review paper that summarizes potential origins of organic Faraday responses. This review (which was already cited in our manuscript) indeed discusses the work of Prasad and co-workers (*Nano Lett.* 2016, 16, 5451–5455) who reported chiral sidechain polymers and suggested B-term contributions. However, it is critical to note that Prasad and co-workers propose the B-term Faraday to arise from a “displacement of electron density along a helical trajectory, which can be viewed as generation of curvilinear microcurrents.”. They arrive at this conclusion using computational modelling of isolated polymer chains in vacuum. They further propose that the Faraday response of their chiral polymeric material originates from a light-induced magnetic moment, which would be opposite for left- and right-handed systems (i.e., a Faraday response that is dependent on chirality). We do not agree with this assignment and believe the enantiomeric response of their system to be erroneous (see further comments above). We instead provide an alternative explanation for the large Faraday response: strong excitonic coupling between adjacent polymer chains, with the strong coupling enabled by the chiral supramolecular assembly. We believe exciton coupling is key to amplification of a B-term like Faraday response; this has not been addressed in previous work.

4) what about rigorous quantum mechanical calculations? This would have been interesting to verify the mechanism for high Faraday rotation in organic materials and verify conclusions made in the manuscript.

We agree that rigorous quantum mechanical calculations would be very interesting. However, it is very challenging to apply this approach to a system like ours, despite our best efforts, as we do not have a molecular-level model of our polymer assembly (see response to Reviewer #1, comment 3). Of course, quantum mechanical calculations on simpler model systems could be considered. Whilst interesting, we do not believe such calculations performed on isolated oligomers in vacuum provide significantly new information for verifying the proposed mechanisms. For example, we have previously used TD-DFT of simplified oligomers to show exciton coupling can amplify the (natural) chiroptical response of conjugated polymer systems (10.1039/d1cc02918e) as found experimentally; but the computational expense and disconnect between the simple model system and the actual material limited the impact and insight of this prior study.

5) can the authors provide a clear structure-property relationship that may help us to design new organic materials for Faraday rotation. What type of molecular structure do we need (conjugation, aromaticity, ...)

We agree that structure-property relationships for organic systems that give rise to strong Faraday would be very useful. An analysis of what is known for organic materials was partly explored in a perspective review article by our team (see Reviewer #2, comment 3). Despite this analysis, it is currently difficult to draw any complete conclusions on similar systems to our own as previous literature has suffered from artefacts and an incomplete set of samples (e.g. left and right-handed, racemic). As such, we suggest that defining more complete structure-property relationships should be a key activity for future research. However, at this point, we can conclude the following, based on the understanding gained in this study. We believe that in lower symmetry organic materials (i.e., those unlikely to show Faraday A or C terms), supramolecular aggregation into defined aggregates with long-range exciton coupling between adjacent chromophores will be important. Chirality is one means to control the robustness and uniformity of such aggregates in the solid state.

Reviewer #3 (Remarks to the Author):

This report describes the magneto-optical activity of chiral supramolecular polymers. They prepare thermally annealed blends of an achiral polyfluorene and a chiral azahelicene, specifically aza[6]H. The results include CD, MCB, Faraday rotation, and absorption data of the pure compounds as well as the thermally-annealed blends. The primary motivation for this publication is the connection between natural optical activity (due to a chiral center) and the magneto-optical response of the material. At present, there is little theoretical connection between these two effects or phenomena. However, there are many in the magneto-optical community, the CISS community and the chiral polymer community that are beginning to acknowledge that there must be a connection. Indeed, the authors invoke an excitonic coupling mechanism to explain their results. The reasoning being that the coupling of electric dipoles must necessarily couple magnetic dipoles; magnetic dipoles associated with an electronic transition are directly measured in magneto-optical spectroscopy, regardless of the technique. Thus, this will be a cutting-edge report when suitable for publication. I have several comments below that must be addressed prior to publication.

We are very grateful for the positive feedback and the praise of our findings and measurements, and we are very happy to address the additional questions.

1. As stated above, all magneto-optical spectra measure the magnetic dipole associated with an electronic transition. The magnetic dipole of an intense electronic transition is larger than that of a weak electronic transition. Figure 3 shows the absorption and Faraday rotation spectra as a function of thickness, but how does the absorption and Faraday rotation change as a function of concentration of the chiral center at a selected thickness, say 333 nm. Was this study performed?

We apologise if we were not clear on the role of the chiral additive in this present manuscript. As mentioned in our response to Reviewer #1, comment 3, we have described the structure and chiroptical properties of this material extensively in previous work (10.1038/s41467-020-19951-y). With respect to the chiral additive (10wt% of aza[6]helicene) specifically, this induces a chiral supramolecular assembly into an otherwise achiral polymer. The resulting absorption, natural optical activity and magnetic optical activity all result from the conjugated polymer assembly, not the additive. This is shown by comparison of the neat aza[6]helicene, achiral polymer and blend chiral polymer circular dichroism spectra as reported in our first paper on these materials (10.1002/adma.201204961). We have since investigated the impact of increasing the chiral additive concentration in achiral polymer blends (10.1039/D1TC05403A) and found that higher wt% of the additive disrupts the molecular packing of the polymer and reduces the chiroptical response. In F8BT specifically, helicene loadings of >10wt% result in phase separation, leading to islands of polycrystalline helicene and bad film quality. On the other hand, loadings <10% result in low chiral induction of the polymer. As such, we are not able to study the concentration of the chiral additive for F8BT.

To assist the reader, we have added an explanation into the supplementary information where we discuss the supramolecular structure of this material:

In these assemblies, the aza[6]helicene additive acts solely as an inducer of the chiral phase into the otherwise achiral polymer. 10wt% of the helicene is an optimum amount of dopant to achieve this good quality films with a large induced chiroptical response (10.1039/D1TC05403A). It is important to note that the measured absorption, natural optical activity and magnetic optical activity originate from the polymer, not the additive.

The Faraday rotation is equivalent to the MORD (Magneto-Optical Rotatory Dispersion) spectrum, which can be modeled by a simple second derivative of the absorption spectrum. Have the authors considered this simple calculation and overlaying it with the Faraday rotation? If the Faraday rotation spectral response and this second derivative deviate from one another as a function of concentration, then this might indicate a synergistic response, or at least other contributions to the Faraday rotation spectrum.

In our revision, we have attempted to model the Faraday rotation spectrum using the second derivative of the absorption, as suggested. The second derivative data, shown below, contains a bisignate peak that is not present in our experimental data. This analysis provides further evidence, as we suggest in our manuscript, that we have overlapping B-term signals that generate a so-called “pseudo A-term” line shape.

Left: Absorption spectrum of the 216 nm thick film of F8BT:aza[6]H-M; Right: Measured Verdet constant (red) and 2nd derivative of the absorbance (blue).

2. The authors clearly state that the thickness of the polymer is critical to the magnitude of the Faraday rotation. I believe that they are obligated to characterize the material more substantively. For example, how do the authors know that the effect is not due to some new liquid crystalline domain or similar phase? This is critical since the report cites a molecular (supramolecular) mechanism and not a material one. I recommend polarized microscopy or some XRD or X ray scattering or similar to help define the material more clearly; of course, other approaches are equally valid. If the sample has created some phase or domains with differing refractive indices, then the absorption will be different and accordingly the magnetic dipoles associated with those domains will be different. Or, thought another way, what is the annealing process doing and what processes are operative that lead to the increase in Faraday rotation as the sample thickness is increased?

We have extensively investigated the impact of thickness of polyfluorene:aza[6]helicene films on their chiroptical and structural properties in [10.14469/hpc/5914](https://doi.org/10.14469/hpc/5914), [10.1038/s41467-020-19951](https://doi.org/10.1038/s41467-020-19951) and [10.1039/d1tc05403a](https://doi.org/10.1039/d1tc05403a). The absorption spectra and cross-polarised optical microscope images show almost no changes that would be associated with a change in phase or structure, and once corrected for reflection, the absorbance normalised circular dichroism (g_{abs}) is constant with respect to thickness. We performed additional GIWAXS and AFM

measurements on as-cast and annealed films of neat and blend (F8BT:aza[M]) thin films, including thicker films of aza[M]. As can be seen from the data below, annealing increases molecular ordering (both lamella and π - π stacking), but the increased order is consistent for neat and blend films, irrespective of film thickness. This data has been added to the Supplementary Information.

Figure S10: GIWAX (first and second row) and AFM (third row) data for as cast (first row) and annealed (second row) neat F8BT (first column) as well as thin and thick F8BT:aza[6]M (second and third column) blends.

Earlier work of Meijer and co-workers on chiral sidechain polyfluorenes (10.1002/adma.200305243) has shown that *very* thin films ($\sim <40$ nm) of this similar material can give rise to different structures when the polymer coverage is not sufficient to cover the entire substrate. None of the samples measured in this work are in that regime however.

Although we have not previously observed structural changes or changes in g_{abs} as a function of film thickness, we have observed a dependence on thickness of the circularly polarised luminescence. The origin of thickness-dependent optical phenomena in this class of materials is still unclear and a subject of ongoing investigation by us and others (10.1039/D1CC02918E, 10.1038/s41598-024-63126-4).

In summary, at this point it is not clear what causes the thickness dependency of the Verdet constant and this does not seem to be caused by clear and gross changes in the structure of the polymer assembly. We can only speculate that it has to do more subtle changes in the

supramolecular ordering in the optical axis. We hope that future studies we will be able to further clarify the origins of this thickness dependency.

With respect to the revised manuscript, we have combined Figure S10 as well as the responses to reviewer #1, comment 3 and reviewer #3, comment 1 and added them to the Supplementary Information section 6. We hope that this discussion clarifies open questions on the role of the supramolecular assembly.

3. The authors report the MCB (Magnetic Circular Birefringence) data and note that "Despite a very different measurement methodology, the obtained data is consistent with significant Faraday rotation..." Of course, the authors know that Magneto-Optical Rotary Dispersion, Magnetic Circular Birefringence and Faraday rotation all measure the same magnetic permeability of the sample. The way in which this section is written almost implies that these are different quantities. I urge the authors to write this in more clear fashion.

We apologise for the confusion we have caused. We have adjusted the relevant text in the paper, emphasising that MCB and Faraday rotation are indeed the same physical phenomenon and that only the measurement set-up to obtain this quantity is different:

Magnetic circular birefringence (MCB), which gives rise to Faraday rotation, was extracted from Mueller-matrix polarimetry performed in a positive and negative applied magnetic field ($B_z = 1.3$ T) and corrected for contributions of the substrate.³⁰ Despite the very different set-up for measuring the same quantity, the results are consistent with significant Faraday rotation at ~500nm (Figure S7).

4. I congratulate the authors for addressing their instrumental artifacts. The balanced detection method is only appropriate for samples exhibiting small Faraday rotations; samples that feature large rotations will show artifacts due to signal processing errors. Some researchers have placed a quarter wave plate retarder after the sample and before the detector to compensate for this. I believe that more discussion of these instrumental details in the literature will help others avoid mistakes.

We highly appreciate the positive comment about our work to address the instrumental artefacts. This was one of the big challenges of this study and became one of the most important findings and take-home messages for all of us. We note that since our submission F. Zinna has also flagged this issue (10.1002/chir.70024) citing our preprint of this paper as an important example of good practice. To amplify the message, as requested, we have now cited the Zinna publication in our manuscript where we describe the artifacts (now reference 29).

5. Magneto-optical activity can be measured by Faraday rotation, MCB, MORD and MCD (Magnetic Circular Dichroism). In fact, the total magneto-optical response of a material is the sum (Faraday rotation or equivalent and MCD), even though a device may operate on a Faraday rotation mechanism. The authors note the interest in porphyrins, but only cite their work (citations 13 and 14). MCD spectra of porphyrins is well studied (for decades and recently) and some other contributors should be listed. For example, the separate work of Stillman and Mack (though they have some collaborative manuscripts) have discussed in detail the magneto-optical properties of porphyrins that should be mentioned. But, one could equally cite Stephens (cited in other contexts), Gouterman, Djerassi, Michl, and many others. While their (collective) primary interest was/is band assignment and electronic structure, their interpretations are applicable to Faraday rotation.

We highly appreciate the vast literature on MCD and apologise if we have underrepresented important contributions from others. One issue we encountered was to try and limit examples to keep the bibliography concise and therefore focused on prior work (e.g., references 13 and 14) specifically focused on Faraday rotation. In our revision, we have included the following additional references (now references 15-19):

Kobayashi, N., Nakai, K. Applications of magnetic circular dichroism spectroscopy to porphyrins and phthalocyanines. *Chem. Commun.* 4077-4092 (2007).

Mack, J., Stillman, M.J., Kobayashi, N. Application of MCD spectroscopy to porphyrinoids. *Coord. Chem. Rev.* **251**,429-453 (2007).

Zhang, A., Kwan, L., Stillman, M.J. The spectroscopic impact of interactions with the four Gouterman orbitals from peripheral decoration of porphyrins with simple electron withdrawing and donating groups. *Org. Biomol. Chem.* **15**,9081-9094 (2017).

Jacek, W., Josef, M. Perimeter model and magnetic circular dichroism of porphyrin analogs. *J. Org. Chem.* **56**,2729-2735 (1991).

Keegan, J.D., Bunnenberg, E., Carl, D. Magnetic circular dichroism studies - 63. Sign variation in the magnetic circular dichroism spectra of some perimeter symmetric metallo porphyrins. *Spectrochim. Acta A* **40**,287-297 (1984).

6. Faraday B-terms typically show peak maxima (or minima, if negative) that coincide with the absorption maxima. In all of the spectra shown here, there appears to be a measurable shift to lower energy of the Faraday rotation peak minima (these are negative peaks) relative to the absorption maxima. Do the authors have insight into this observation? Is this due to some instrumental feature (does the Wollaston prism have a poor spectral response here), or is this due to the presence of a positive B-term on the high energy side that is smaller in magnitude?

In MCD, the maximum rotation in a standard B-term signal coincides with the absorption maximum. In contrast, Faraday rotation (or MCB) spectra that originate from a standard B-term display a bisignate signal that inverts sign at the absorption maximum. Hence, the wavelength of maximum rotation is offset relative to the maximum absorbance wavelength. In our case, two overlapping B-terms, which combine to give a pseudo A-term line shape, complicate the analysis. Nonetheless, each overlapping B term is shifted with respect to one another, creating a pseudo A-term signal with a maximum shifted (slightly) from the absorption maximum. If the overlapping B terms were not shifted with respect to one another, they would simply cancel each other out. This is why the wavelength of maximum rotation is slightly shifted away from the absorbance maximum. We do not believe that the shift is caused by dispersion of the Wollaston prism, as this should affect absorbance and Faraday rotation in the same way.

Author correction:

During the review process we realized there was an important typo in our original submission: we missed the inclusion of a "shared authorship note". Leo Delage-Laurin, David Reger and Abdusalom A. Suleymanov all contributed equally to this work. This has been corrected in the revision.

Additionally, the University of Oxford was added as an affiliation for David R.

Additional changes:

We have made some minor changes to the text which are all marked in red as well.